# Mutual Augmentation of Spectral Sensing and Machine Learning for Non-Invasive Detection of Apple Fruit Damages

**Boris Shurygin** [1,2,*], **Igor Smirnov** [3], **Andrey Chilikin** [3], **Dmitry Khort** [3], **Alexey Kutyrev** [3], **Svetlana Zhukovskaya** [4] **and Alexei Solovchenko** [1,2,3,*]

1. Faculty of Biology, Lomonosov Moscow State University, 119234 Moscow, Russia
2. School "Brain, Cognitive Systems, Artificial Intelligence", Lomonosov Moscow State University, 119234 Moscow, Russia
3. Federal Scientific Agroengineering Center VIM, 109428 Moscow, Russia
4. Michurin Federal Scientific Center, 393766 Michurinsk, Russia
* Correspondence: lodinn@lodinn.com (B.S.); solovchenkoae@my.msu.ru (A.S.); Tel.: +7-(495)-9392587

**Abstract:** Non-invasive techniques for the detection of apple fruit damages are central to the correct operation of sorting lines ensuring storability of the collected fruit batches. The choice of optimal method of fruit imaging and efficient image processing method is still a subject of debate. Here, we have dissected the information content of hyperspectral images focusing on either spectral component, spatial component, or both. We have employed random forest (RF) classifiers using different parameters as inputs: reflectance spectra, vegetation indices (VIs), and spatial texture descriptors (local binary patterns, or LBP), comparing their performance in the task of damage detection in apple fruit. The amount of information in raw hypercubes was found to be over an order of magnitude excessive for the end-to-end problem of classification. Converting spectra to vegetation indices has resulted in a 60-fold compression with no significant loss of information relevant for phenotyping and more robust performance with respect to varying illumination conditions. We concluded that the advanced machine learning approaches could be more efficient if complemented by spectral information about the objects in question. We discuss the potential advantages and pitfalls of the different approaches to the machine learning-based processing of hyperspectral data for fruit grading.

**Keywords:** fruit grading; machine learning; image processing; hyperspectral imaging; object classification

## 1. Introduction

Non-invasive plant phenotyping is a blooming area of research with broad applications in the agriculture, including horticulture. It is the basis of cultivar performance monitoring in the field and, more specifically, a proxy of fruit yield, ripeness, soundness, and storability [1–4]. Since "wet" biochemical analyses of fruits are laborious, expensive, and in some cases hardly feasible, non-invasive approaches to retrieval ripening-related changes in the pigment composition, presence of mechanical damages, physiological disorders, and lesions by phytopathogens on the analysis of reflected light spectra have been suggested [5,6]. Many of the underlying parameters were initially monitored by visual observations and scoring [7,8], but these approaches are rapidly replaced by automated image analysis based on computer vision [9–13], especially in sorting line equipment.

Rapid advances in computer vision over the past decade have resulted in unprecedented increase in precision and overall performance of machine learning (ML) models, now rivaling that of human experts for certain problems [14]. This has proven to be a mixed blessing, however, as most of this progress was achieved by using inherently poorly interpretable, black-box approaches, which are limited in their applicability in fully automated control scenarios and further give rise to ethical problems [15–17]. Furthermore, while promising spectacular results, these techniques introduce an issue of knowledge

retention—most of the time, every new generation of deep learning models marks a new beginning for their employment in specific domains, meaning large portions of earlier work end up being discarded.

On the other end of the scale reside "classic" methods. They are grounded in fundamental biophysical properties of plants and aimed at uncovering relationships between the physiological state and its manifestations through non-invasively observable variables. Of particular note is spectroscopy, the approach spanning at least several decades of plant research [18,19]. During that time, a wealth of knowledge about pigments, plant interactions with light, and non-destructive phenology assessment has been accumulated. Specific examples include reflectance-based monitoring changes in pigment composition induced by ripening, environmental stresses, and phytopathogen attacks [20–24]. One of the most simple, yet powerful techniques emerging from this development is vegetation indices (VI), transforming quantification of incident radiation into highly biologically relevant parameters such as nitrogen and water content, pigment pools, and overall biomass [25].

Initially starting with point measurements, plant spectroscopy has since expanded into imaging technologies, which are especially relevant to horticulture as opposed to field crops management [1], as the heterogeneity of plant organs and tissues makes bulk assessments unfeasible [22,26]. With the advent of accessible imaging spectrometers, and in the context of a long-standing issue of impracticality in obtaining and processing hyperspectral images as compared to multispectral ones, there are ongoing debates about both approaches to spectral image processing and designing spectral imagers for purposes of remote sensing of plants. In particular, high dimensionality of hyperspectral data imposes significant restrictions on deep learning architectures stemming from the computational complexity. Learning spectral features comes at a cost of a reduced efficiency in processing spatial information, and knowledge-based dimensionality reduction could increase phenotyping throughput, simplify hardware operations, and allow for larger and more complex deep learning architectures to be used.

In this work, we aimed at dissecting the information content of hyperspectral images for non-invasive assessment of fruit health focusing either on spectral component, spatial component, or both. To this end, we have employed popular "classic" machine learning approaches: random forest (RF) and support vector classifiers (SVC) and compared their end-to-end performance on a relatively large hyperspectral image dataset as a measure of the information retention after dimensionality reduction was performed. These approaches were chosen in large part owing to their interpretability: while current and future deep learning architectures are likely to boast higher absolute performance, they also imminently link it with a choice of architecture and a wide range of hyperparameters, complicating comparative analysis. Additionally, we strived to find an approach potentially usable in real-life fruit sorters, which have to process many dozens of fruits per second. In effect, this means working in real-time, so the lightweight approach is of utter importance. For line scanning cameras, pixel-based classification techniques have the benefit of avoiding computationally expensive preprocessing related to geometric correction of the images. Although RF classifiers were notably utilized in a recent work [27] concerning early bruise detection in apples, we have extended this approach by simulating different spectral resolutions of the image sensor and comparatively investigated the performance of vegetation indices (VI) and spatial texture descriptors (local binary patterns, LBP [28]) as our main goals.

## 2. Materials and Methods

### 2.1. Plant Material

Fruit of apple (*Malus × domestica* Borkh.) (*n* = 100) variety "Gala" (red–green-colored) were grown in a commercial orchard of "Zorinsky Sad" fruit growing company (Oboyan, Kursk region, Russia). The fruit were hand-picked from the tree and stored in a conventional atmosphere until measured. The fruit were imaged within one week from picking from the tree. The fruit lacking visually discernible symptoms of damage were selected to-

gether with those bearing the symptoms of various damages, including sunburn, sunscald, fungal lesions, and mechanical damages and bruises (see Figures S1 and S2).

## 2.2. Setup for Hyperspectral Measurements

The hyperspectral reflectance images of the apple fruits were taken with an in-house-made measurement setup mimicking a conveyor sorting device (Figure 1) fitted with a scanning (push-broom) imaging hyperspectrometer BaySpec OCI-F (Bayspec Inc., San Jose, CA, USA). For each pixel of a hyperspectral image, a reflectance spectrum (spectral range 400–1000 nm; spectral resolution 5–7 nm; 512 pixels/line) was recorded against a reflectivity standard made of Spectralon. The camera was externally controlled, and the raw data were pre-processed by the companion software SpecGrabber and CubeCreator. Four tungsten-halogen lamps (Camelion GU10 35W) assembled in a mount were used as a light source for the apple fruit imaging.

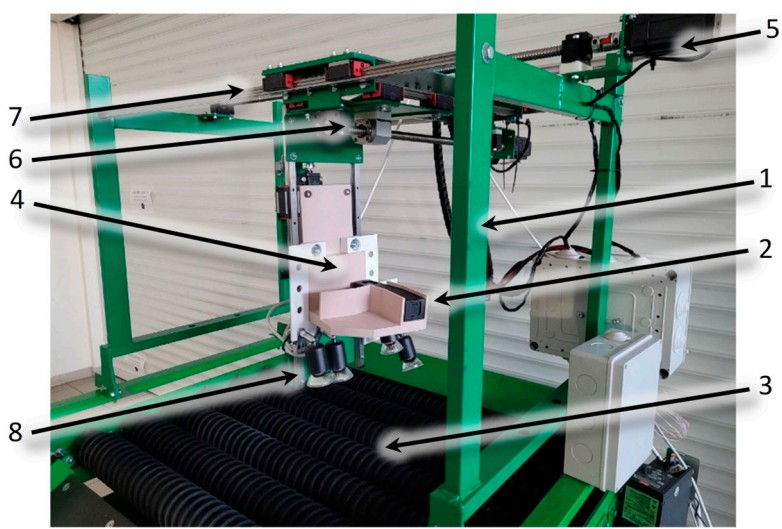

**Figure 1.** The setup for the hyperspectral imaging of apple fruits. Designations: chassis (1); the hyperspectral camera (2), a table with rubber rollers (3) camera suspension (4), stepper motors (5, 6) and the transmission (7), light source (8).

## 2.3. Hyperspectral Data Treatment

### 2.3.1. Expert Annotation (Ground Truth)

Hyperspectral images were converted to corresponding color representations using an approximation of CIE 1964 observer spectral response curves and with D65 standard illuminant [29,30]. Two human experts performed annotation with bit masks using the Supervisely online labeling tool [31], assigning classes "Fruit" for intact tissues and "Damage" for visibly damaged regions of the apple surface. Unlabeled parts of the image were considered as background. Per-pixel Cohen's kappa [32] between human expert annotations for every image was $0.93 \pm 0.05$, establishing the model performance baseline. Unless otherwise stated, the results are presented for one of the annotations used for both training and validation; the other annotation was used to test the classifier stability with respect to the training labels used.

### 2.3.2. Per-Pixel Classification

Per-pixel classification was performed by Random Forest and Support Vector classifiers using scikit-learn Python package [33]. The entire dataset was then split in an 80:20 ratio between training and test image sets. In each set, every pixel was first represented by a feature vector, consisting of either spectral reflectances or VIs and, if applicable, LBP values, and ground truth feature vectors were extracted from each image. Classifiers were trained on a small subset (up to 3000 pixels per image, less than 1% of the overall pixel count) of the resulting feature vectors. Then, with the same transformations applied to the

entire dataset, inferences were drawn for all pixels, and the results were compared to the ground truth (expert annotation).

Decision trees are known for being sensitive to the class balance; thus, two strategies were used: extraction of an equivalent number of pixels for every class present in the image (oversampling of the minority class) or proportionally to the relative abundance of a given class on that image. Corresponding features were drawn randomly without replacement from pixels of their respective classes. Since the number of samples used for training was low, classifier performance was evaluated for the entire dataset, with separate figures reported for the test set performance only. The first six spectral channels were excluded from the subsequent analysis due to their high noise. Shannon entropy was used as a criterion for node-splitting, and a default maximum number of features under consideration equal to the square root of the total number of input features was used. A preliminary search for the optimal number of estimators and tree depth limit was performed for non-downsampled features and transformed features, with the number of estimators ranging from 100 to 500 in steps of 100, and the maximum tree depth assuming values from the list (5, 6, 7, 8, 9, 10, 12, 14,1 16, 18, 20, 25, 30). The value of $n$ = 300 trees in the forest was set for each classifier. For non-transformed input features, the maximum tree depth was set at 10, as higher values indicated overfitting on the test set (not shown), and for transformed features, it was increased to 20. For the proportional sampling, an application of class weights was also tested.

Support Vector classifiers were trained using the LinearSVC implementation with hinge loss, one-vs-rest multi-class optimization strategy, and default regularization.

For classifiers operating directly on spectral reflectances, coarser spectral resolution was modeled by linear downsampling of input hypercubes along the spectral axis by factors of 1 (non-modified), 1/2, 1/4, 1/8, 1/16, and 1/32.

### 2.3.3. Vegetation Indices as Feature Transformations

For this analysis, we have selected vegetation indices representing the main pigment groups in fruits: a sensitive indicator of chlorophyll (Chl) content, $CI_{700}$ (commonly also called Simple Ratio, $SR_{700}$) [20,25,34], mARI [35], an index tightly related with anthocyanin (AnC) content of fruit [19,36], and mBRI, recently introduced by us as an indicator of carotenoid content weakly affected by AnC [22]. Another extremely important addition to this list was the measurement of relative brightness using spectral reflectance in a near-infrared (NIR) band unaffected by pigment absorption of light (e.g., $R_{800}$). This parameter often gets overlooked, however; while spectral indices are specifically designed to be minimally affected by changes in illumination conditions, spatial distribution of apparent brightness contains information about the object geometry. Definitions of the indices used are presented in Table 1.

**Table 1.** Vegetation indices used as feature transforms.

| Index Name | Formula | Explanation |
|---|---|---|
| NIR band | $R_{800}$ | Highly invariable in heathy plant tissues and affected by damages, it also contains information about the viewing geometry and illumination |
| $CI_{700}$ | $CI_{700} = \dfrac{R_{800}}{R_{700}}$ | $R_{800}$ is unaffected by the pigment absorption of light, whereas $R_{700}$ corresponds to the Red Edge region of the red Chl absorption maximum |
| mARI | $mARI = \dfrac{R_{800}}{R_{550}} - \dfrac{R_{800}}{R_{700}}$ | $R_{550}$ is affected by both AnC and Chl, and $R_{700}$ is the reflectance in the band of the red Chl absorption maximum. |
| mBRI | $mBRI = \dfrac{1}{R_{640}} + \dfrac{1}{R_{800}} - \dfrac{1}{R_{678}}$ | $R_{800}$ and $R_{640}$ are used as the terms sensitive to the accumulation of the damage-related pigments and reflectance $R_{678}$ is employed for correction of the index for the interference from Chl absorption |

### 2.3.4. Combined Spectral–Spatial Classification

Multi-scale local binary patterns (LBP) were used for the embedding of spatial features. They were calculated by means of scikit-image [37] Python package. For non-transformed features, the average of all spectral channel reflectance was used as the input graylevel image; for transformed features, the NIR reflectance band was used. In all cases, eight points were used for quantization of the angular space, rotation invariant implementation was chosen, and six different spatial resolutions of the operator were used: 4, 8, 16, 32, 64, and 128 pixels, respectively. The overall resulting pipeline is shown on Figure 2.

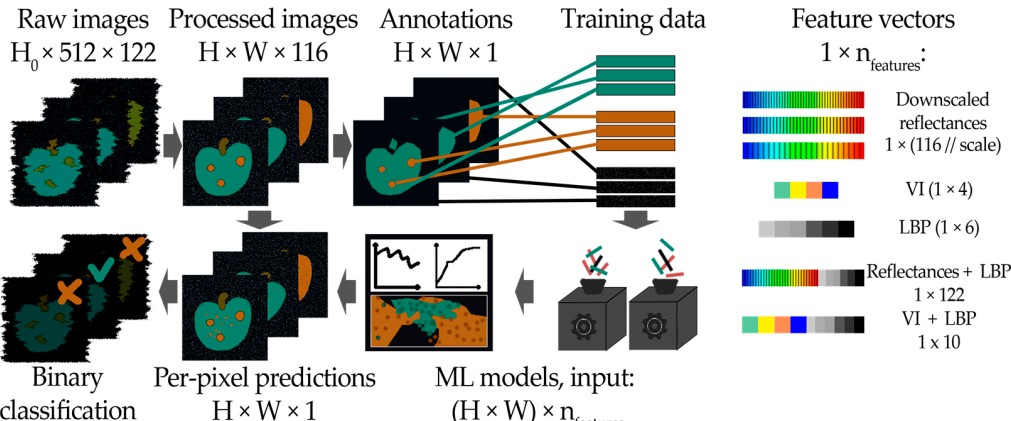

**Figure 2.** An overview of the classification pipeline. After preprocessing and image registration, both spatial dimensions of spectral images vary; one-hot label encoding is, in effect, used internally during training and inference.

### 2.4. Performance Evaluation

Per-pixel classification results were taken as an intermediary output and compared to ground truth masks. Accuracy, Cohen's kappa (see above) and F-score for damaged fruit tissue with beta = 2 (placing a greater emphasis on recall than precision) were calculated on a per-image basis, and their averages and standard deviations reported. For F-score, the zero-division case (no pixels assigned to the "Damages" class in ground truth nor resultant classification) were ignored during averaging. Per-image ratios of the total number of pixels classified as damage to the total number of pixels classified as fruit (including damages) were used as a metric to perform the final "damaged-intact" fruit classification. Receiver operating characteristics (ROCs) and precision-recall curves were calculated for each classifier, with the minority class being damaged fruits; optimal detection thresholds were determined using the $F_2$-score.

## 3. Results

The overall per-pixel classification accuracy for most models varied between 94.6% and 98.3% (Table 2). In some cases, it exceeded the inter-expert agreement value of 97.7%, making higher accuracy values not necessarily indicative of a better performance. These results, however, were stable with respect to label noise: using alternative ground truth labels for validation has resulted in a reduction of overall accuracy by approximately 1% across the board, but the relative performances of classifiers did not change significantly (Table S2). Similarly, using a different annotation for training resulted in high average pairwise inter-classifier kappa agreement of 0.99 ± 0.01.

**Table 2.** Per-pixel classification results.

| Feature Set Used | Accuracy, % | | Cohen's Kappa | F$_2$ Score |
|---|---|---|---|---|
| Baseline (agreement between human experts) | 97.7 ± 2.4 | | 0.931 ± 0.053 | 0.445 ± 0.309 |
| Random Forest classifiers | | | | |
| Reflectances with spectral downsampling | 1 [1] | 98.2 ± 2.2 | 0.944 ± 0.056 | 0.160 ± 0.286 |
| | 1/2 | 98.1 ± 2.2 | 0.943 ± 0.057 | 0.154 ± 0.280 |
| | 1/4 | 98.1 ± 2.2 | 0.942 ± 0.058 | 0.149 ± 0.275 |
| | 1/8 | 98.1 ± 2.3 | 0.941 ± 0.059 | 0.135 ± 0.262 |
| | 1/16 | 97.8 ± 2.5 | 0.932 ± 0.065 | 0.095 ± 0.212 |
| | 1/32 | 97.8 ± 2.5 | 0.931 ± 0.064 | 0.127 ± 0.233 |
| Reflectances with no downsampling + LBP | 98.1 ± 2.3 | | 0.941 ± 0.058 | 0.173 ± 0.290 |
| Reflectances + LBP + weighting | 96.1 ± 3.6 | | 0.888 ± 0.073 | 0.181 ± 0.241 |
| LBP only | 90.1 ± 4.7 | | 0.651 ± 0.101 | 0.000 ± 0.000 [2] |
| VI only | 98.0 ± 2.1 | | 0.940 ± 0.053 | 0.149 ± 0.266 |
| VI + LBP | **98.3 ± 2.1** | | **0.948 ± 0.054** | 0.192 ± 0.295 |
| VI + LBP + weighting | 98.2 ± 2.0 | | 0.947 ± 0.052 | **0.196 ± 0.294** |
| Support Vector classifiers | | | | |
| Reflectances with spectral downsampling | 1 | **97.9 ± 2.5** | 0.935 ± 0.063 | 0.142 ± 0.260 |
| | 1/2 | 97.8 ± 2.6 | 0.933 ± 0.066 | 0.110 ± 0.228 |
| | 1/4 | 97.8 ± 2.6 | 0.931 ± 0.067 | 0.096 ± 0.213 |
| | 1/8 | 97.7 ± 2.6 | 0.930 ± 0.068 | 0.086 ± 0.204 |
| | 1/16 | 97.5 ± 2.7 | 0.920 ± 0.070 | 0.035 ± 0.131 |
| | 1/32 | 97.4 ± 2.7 | 0.919 ± 0.071 | 0.018 ± 0.083 |
| Reflectances with no downsampling + LBP | **97.9 ± 2.5** | | **0.936 ± 0.063** | 0.140 ± 0.256 |
| Reflectances + LBP + weighting | 95.7 ± 2.1 | | 0.864 ± 0.050 | **0.200 ± 0.273** |
| LBP only | 39.1 ± 4.0 | | −0.105 ± 0.027 | 0.011 ± 0.019 |
| VI only | 97.5 ± 2.7 | | 0.922 ± 0.071 | 0.006 ± 0.039 |
| VI + LBP | 96.8 ± 3.1 | | 0.902 ± 0.075 | 0.029 ± 0.052 |
| VI + LBP + weighting | 94.6 ± 3.1 | | 0.832 ± 0.061 | 0.079 ± 0.128 |

[1] Spectral downsampling factors, 1 corresponds to no downsampling applied. [2] LBP-only RF classifier has not produced any "Damaged" labels.

Random forest classifier using VI + LBP was found to be the most sensitive and accurate, followed closely by the one using raw reflectance values with no spectral downsampling. The performance of SVC and RF classifiers was close, but SVC ones remarkably produced better results using non-transformed features compared to using VIs as inputs and were substantially more sensitive to spectral downsampling, with F$_2$ scores steadily dropping as spectral resolution decreased, which was not observed for RF classifiers. The effect of the reduction of the spectral resolution by the downsampling of RF classifiers was found to be insignificant until the simulated spectral channel became 36 nm wide. Weighting has improved F-scores at the cost of accuracy and kappa, and RF classifiers' accuracy was less affected by it than SVC.

An example visualization of features used for vegetation indices and texture classification is shown on Figure 3.

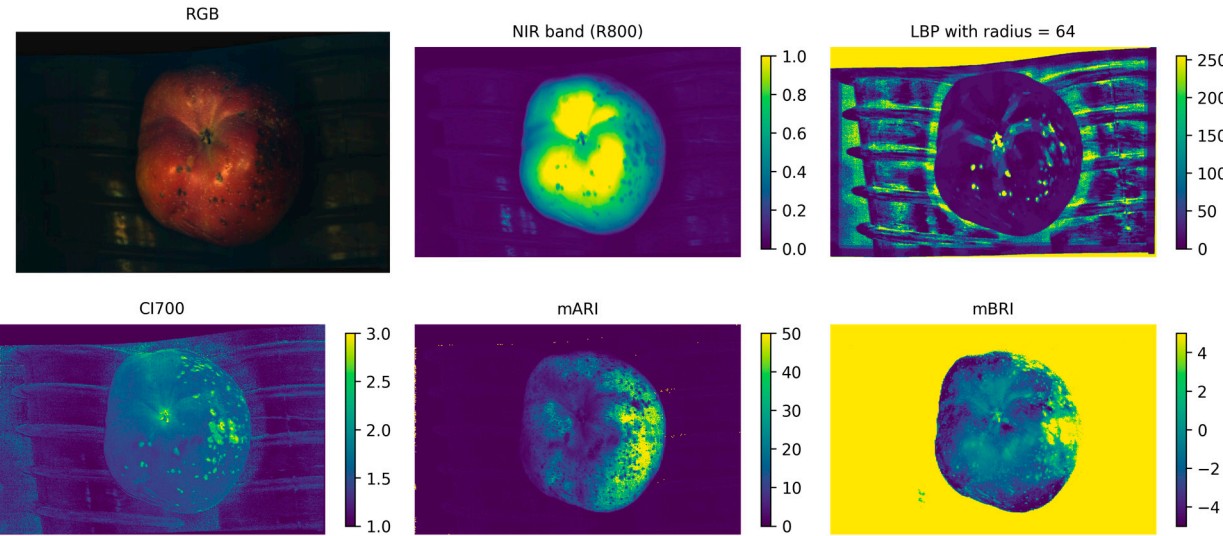

**Figure 3.** Visual representation of some of the features (indicated above the figures) used for the classification of sound and damaged fruit regions (see also Methods).

Representative classification results for different fruit classes and damage types are demonstrated in Figures 4 and 5.

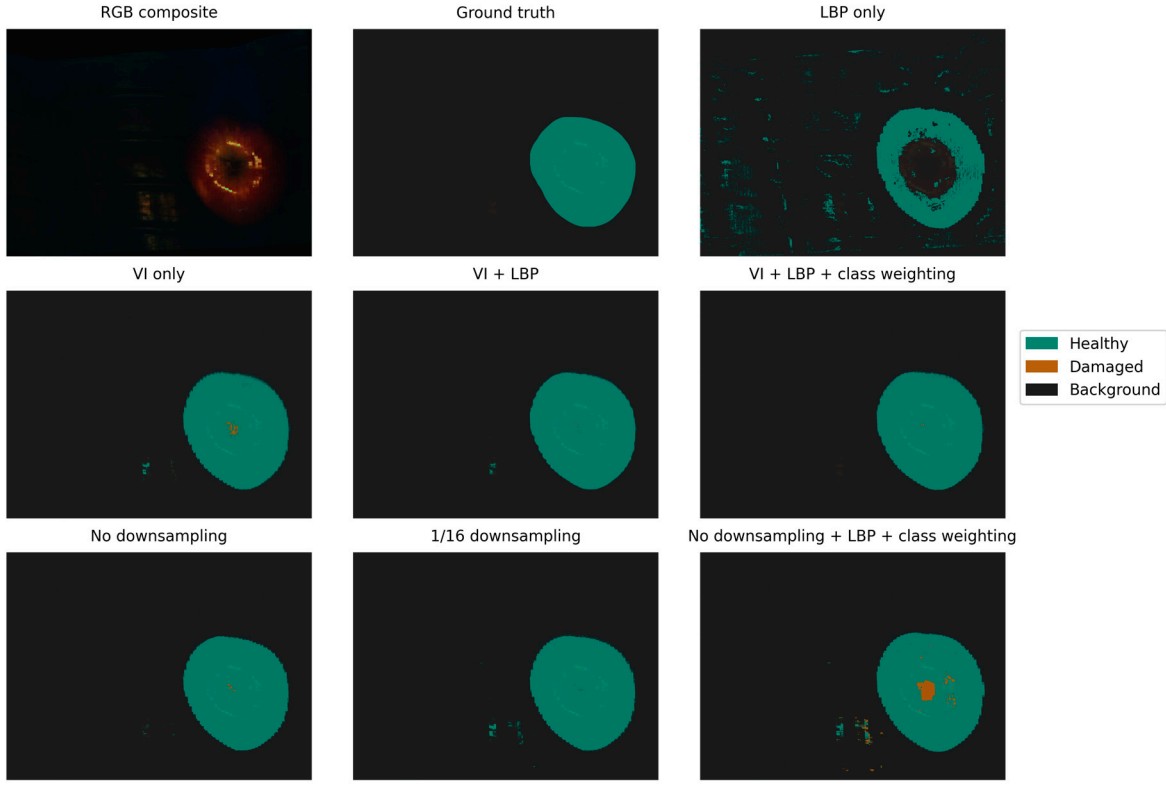

**Figure 4.** Classification results for a healthy apple.

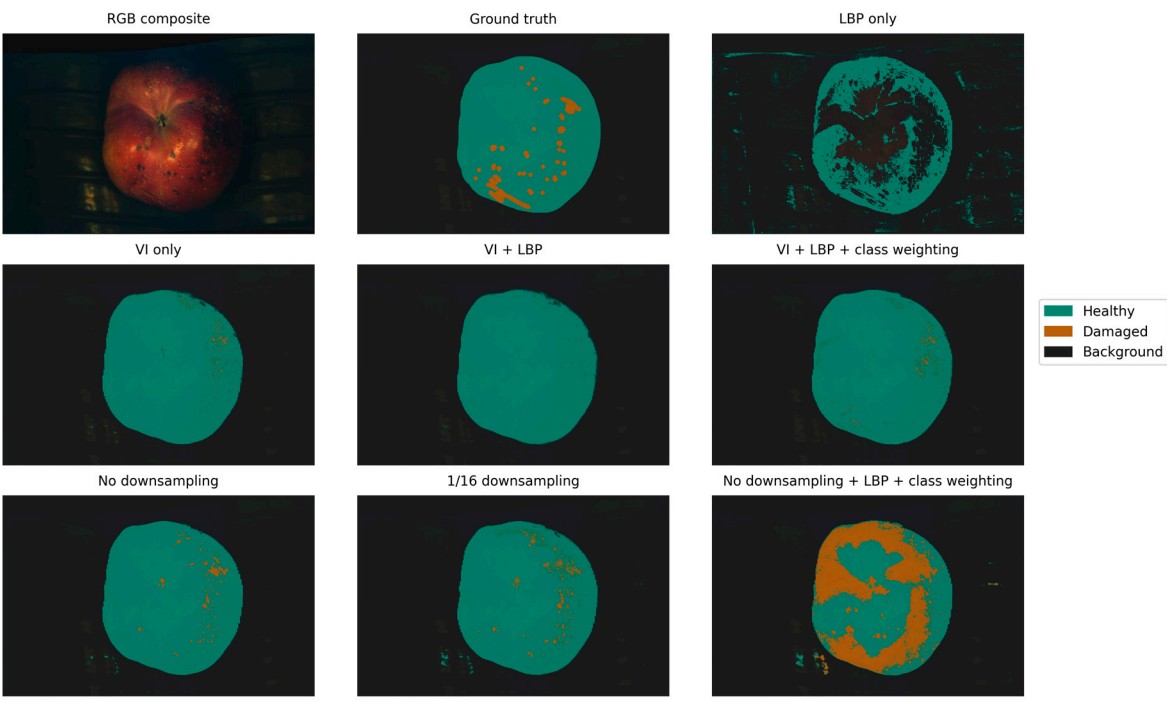

**Figure 5.** Classification results for an apple with fungal lesions.

The texture-only classifier using multiscale LBP turned out to be unusable for damage detection in apple fruit, as it has not classified any pixels as damaged tissues in the entire dataset. The inclusion of spectral features was found to be beneficial. Between the spectral classifiers, the difference in results is small, which is particularly remarkable with respect to the spectral downsampling performed. A vegetation indices-based classifier can be seen to be more resilient to non-uniform illumination conditions (Figures 5 and 6) as compared to the spectral reflectance-based one, but it was less efficient in detection of small damages.

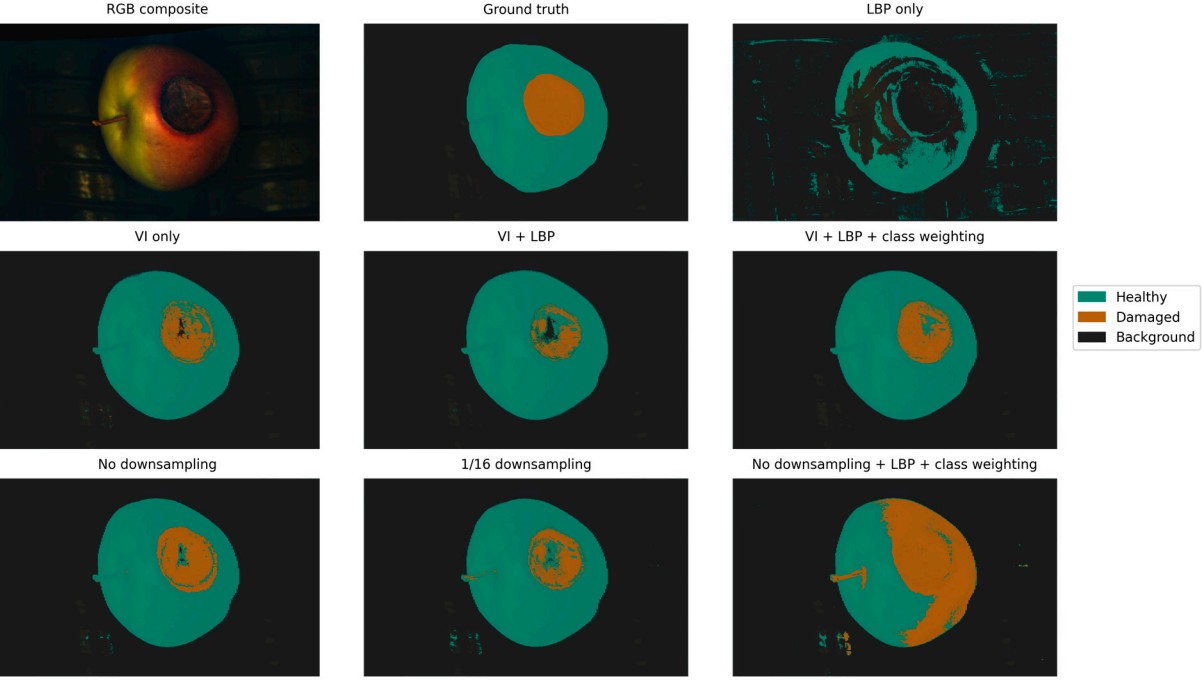

**Figure 6.** Classification results for an apple with a severe sunburn.

Low average $F_2$ scores for the "damaged" class and high variance thereof were indicative of a commonly observed pattern of classifiers either detecting corresponding regions with high confidence or missing them entirely. Images featuring mechanical damages with no prominent tissue degradation were the main contributor to these low scores, with another major factor being detections assuming salt-and-pepper noise-like appearance, thus reducing the recall.

For the end-to-end task of binary fruit classification based on the detected fraction of the damaged surface of the fruit, we have utilized RF classifiers. AUC (area under curve, an efficiency measure commonly used in ML) for the classifiers with no feature transformation was between 0.69 and 0.72, improved to 0.8 for spectral reflectances with no downsampling + LBP + class weighting. VI-only classifier has shown AUC of 0.74, while the addition of LBP and both LBP and class weighting has improved this result to 0.78 and 0.8, respectively. Finally, we have analyzed precision-recall curves and calculated precision and recall at a damage fraction threshold corresponding to the maximum of $F_2$ score (Figure 7). This maximum corresponded to high recall values in all cases, but precision varied between 0.6 and 0.7, with VI-based classifiers having better performance overall. Classifiers operating on spectral reflectances, however, benefitted more from the addition of LBP features and class weighting; however, as can be seen from Figures 5 and 6, this was achieved by overly aggressive damage detection.

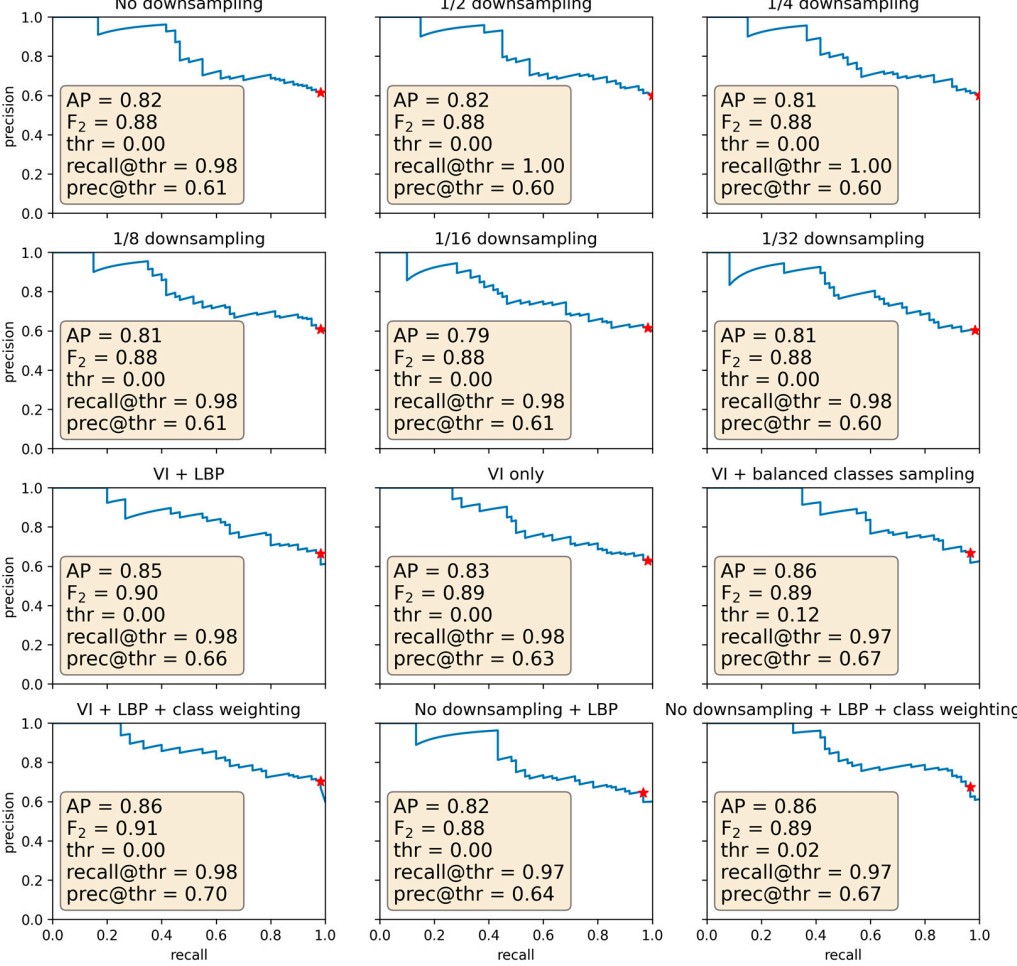

**Figure 7.** Precision-recall curves for different classifiers (indicated above the figures). Inset shows values for Average Precision (AP), $F_2$ score, optimal damage fraction threshold, recall, and precision values at that threshold (corresponding point on the curve marked as red ☆).

## 4. Discussion

In our study, we have tested different approaches to the processing of hyperspectral reflectance data for the detection of damaged regions of apple fruit using an ML-based approach. Features of a different nature (purely spectral such as VIs, purely spatial such as LBP or their combinations) were assessed in terms of RF and SVC classification efficiency. In addition to this, the effect of coarser spectral resolutions on the classification efficiency was estimated.

It is evident from per-pixel classification results (Table 2) that the non-invasive detection of fruit damages using hyperspectral imagery is possible with accuracy comparable to that of human experts. However, both visual inspection and F-score illustrate imperfect alignment of ML results with human annotations, which is one of the major issues impeding progress in automatization [16]. Class weighting and sampling were seen to have an enormous impact on the per-pixel classification performance, and in computer vision applications for plant phenotyping, including assessment of fruit quality, it is crucial to control for the class balance and sampling strategy when evaluating the model. Ultimately, the assessment of the classification performance should be brought into a broader context of potential industrial applications, and the inherent asymmetry in the cost of misclassification addressed (see e.g., [38]). It is important that the best-performing classifier based on the commonly reported accuracy or Cohen's kappa scores might not have the best performance once the full scope of a binary classification problem is considered. Tasks of anomaly detection are associated with highly imbalanced labels, and fruit health assessment is no exception; this consideration makes accuracy a potentially misleading metric. For example, a texture-only RF classifier we have tested in this work produced no labels corresponding to damaged tissues, yet reported an accuracy exceeding 90%, which might be considered high in a broader context with more balanced labels.

All classifiers considered in this work invariably struggled with identifying mechanical damages. Spectral signatures of these areas suggest that, in most cases, no pigment degradation takes place, at least for several days, and the apparent darker coloration is the result of a change in lighting geometry. These depressed regions are indistinguishable from the areas otherwise being in shadow in our imaging setup. Diffuse illumination which is normally desirable to avoid specular reflectance of fruit would also make these damages less apparent. On the whole, this class of damages should be treated separately from more severe damage types such as sunscald, sunburn, or fungal lesions. It is unclear whether such mechanically damages damaged fruits pose a serious risk of spoiling in storage, or whether the suitability of such fruit for long-term storage is more relevant to the quality standards as contracted by the grower and/or the packinghouse. Still, secondary infection, which is unobservable at first for several days but can proliferate and show up at a later stage, is quite possible in the bruised fruit.

It could be seen from Figures 5 and 6 that LBPs and $CI_{700}$ can highlight small lesions; however, they also pick up irregularities in the background, and the final classification better aligns with mBRI rather than with CI. Still, one can expect this given that the small damages do not always trigger a rapid Chl degradation.

Proportional sampling of pixels has resulted in a significantly higher classification accuracy compared to sampling from each class equally. This effect can be attributed to classifiers using the latter very aggressively, shifting the overall balance towards the minority classes. Correspondingly, the sampling proportion could be used to control this behavior, and a more desirable balance could be achieved. This is an important consideration given the imperfect inter-subjective alignment between human experts: one of the significant problems in machine learning is the extraction of latent knowledge driving human decisions in low confidence scenarios. The problem of noisy labels is often being overlooked in machine learning. A clearer delineation of shortcomings of an approach tested from common disagreements between human experts requires multiple annotations, which are rarely conducted, but would pave the way for more efficient employment of techniques such as residual learning.

Apart from the assessment of the potential of the ML-augmented hyperspectral technique suggested here for damage detection, our findings are relevant for another problem which is still very topical. Namely, does hyperspectral imaging have real advantages over conventional RGB imaging for the detection of visual symptoms of damages to plants (apple fruit in our case)? On the one hand, hyperspectral images are extremely rich in information, and always allow an RGB representation of an object to be reconstructed. On the other hand, the incredible level of detail provided by hyperspectral imagers is a trade-off for lower throughput, spatial and/or radiometric resolution as compared to conventional and even multispectral imaging (for an analysis of this tradeoff, see [39]).

The information content of spectral images is closely related to the notion of degrees of freedom (DoF), and it is generally understood that the number of independent variables determining reflectance spectra of vegetation is notably lower than the number of spectral bands. Therefore, a significant reduction of dimensionality is achievable by careful selection of bands; and this is precisely the problem that vegetation indices are designed to solve [40].

Spectral reflectance coefficients at densely positioned wavelengths contain redundant information and relatively few important (from the standpoint of the goal of the analysis) features, and embedding is required prior to feeding them to a deep convolutional network [41–43]. In other words, dimensionality reduction has to be performed either as a part of a neural network architecture (learned features) or in a supervised manner. The latter has distinct advantages of robustness and knowledge transfer between different generations of deep learning architectures. Usage of vegetation indices as feature transforms has been proposed by researchers before, but, to our knowledge, no systematic study of their usage in proximal sensing of vegetation exists.

All this makes hyperspectral imaging somewhat impractical in the industrial applications. However, it remains a perfect research tool for knowledge-based design of spectroscopic techniques for plant phenotyping, allowing for robust optical monitoring of changes in pigment and nutrient content and offering considerations for industrial imager design suitable for high-throughput fruit sorting lines.

## 5. Conclusions

In this research, we have outlined an end-to-end approach to the fruit classification problem by means of computer vision and identified a number of potential issues and pitfalls. We have considered computer vision and ML applications for the problem of postharvest fruit sorting, and have found a pronounced alignment problem: namely, high per-pixel classification accuracies can be achieved, but these results do not translate well into the problem of fruit grading.

Remarkably, the amount of information in raw hypercubes was found to be hugely (by over an order of magnitude) excessive for the end-to-end problem of classification. Converting spectra to vegetation indices has resulted in a 60-fold compression with no significant loss of information relevant for phenotyping and more robust performance with respect to varying illumination conditions.

At the same time, it became obvious that even the advanced machine learning approaches could be more efficient if they are complemented by spectral information about the objects in question. As a result, a balanced approach to obtaining the image data for computer vision-based fruit grading seems to be most productive and cost-effective. For the implementation of such an approach, capturing images at a few carefully selected spectral channels would be sufficient, although one should be aware of the limitations outlined above. A knowledge-based dimensionality reduction would drastically shorten the adoption time for the newest "mainstream" deep learning architectures to spectral proximal sensing, removing the need to compromise on the learning of spatial features. Powerful but costly and relatively slow hyperspectral sensors will occupy the R&D niche for the development of novel and improved non-invasive methods of the assessment of fruit quality.

**Supplementary Materials:** The following supporting information can be downloaded at: https://www.mdpi.com/article/10.3390/horticulturae8121111/s1, Figure S1: Representative photos of the studied apple fruits (conventional RGB images); Figure S2: Representative RGB images of the studied apple fruits (composed from their hyperspectral images) with a typical reflectance spectrum (shown in overlay) of the damaged area; Table S1: Per-pixel classification results, inference on the test set only; Table S2: Per-pixel classification results using annotations from human expert #2.

**Author Contributions:** Conceptualization, A.S. and B.S.; methodology, B.S. and D.K.; software, B.S. and A.C.; validation, B.S. and A.S.; formal analysis, A.C.; investigation, A.S. and A.K.; resources, S.Z.; data curation, A.S.; writing—original draft preparation, B.S.; writing—review and editing, B.S., A.S., I.S., and D.K.; visualization, B.S.; supervision, A.S.; funding acquisition, I.S. All authors have read and agreed to the published version of the manuscript.

**Funding:** This research was supported by a grant of the Ministry of Science and Higher Education of the Russian Federation for large scientific projects in priority areas of scientific and technological development (grant number 075-15-2020-774).

**Institutional Review Board Statement:** Not applicable.

**Informed Consent Statement:** Not applicable.

**Data Availability Statement:** The data are available from the corresponding author on a reasonable request.

**Conflicts of Interest:** The authors declare no conflict of interest.

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
