# Peer review of "Mutual Augmentation of Spectral Sensing and Machine Learning for Non-Invasive Detection of Apple Fruit Damages"

_horticulturae, doi:10.3390/horticulturae8121111_

Round 1

Reviewer 1 Report

This paper proposed a method for apple damage detection based on spectral information and machine learning. The following parts need to be improved:

1.     In Fig.1, where is “light source 8”. (line 106)

2.     In Materials and Methods, sample quantity should be supplemented. 

3.     What is the standard of the damage? It is better to supplement some pictures.

4.     Too little information about the machine learning introduction.

5.     In the discussion, different methods for processing hyperspectral reflectance data were mentioned, and it is suggested to supplement them in the test methods.

Author Response

We appreciate the careful analysis and positive evaluation of our manuscript by the reviewer. Please see below our detailed responses to the comments made/questions raised by the reviewer. Specific amendments made to the manuscript are highlighted using Tracking Changes mode in the manuscript itself.

  1. In Fig.1, where is “light source 8”. (line 106)

RESPONSE: the figure annotations were corrected.

  1. In Materials and Methods, sample quantity should be supplemented. 

RESPONSE: the number of samples (fruits picked for the experiment) is specified in the first line of the section 2.1. Plant material.

  1. What is the standard of the damage? It is better to supplement some pictures.

RESPONSE: the representative photographs reflecting the typical view of the damages are shown in Figs. S1 and S2 in the online supplementary. These figures are referenced in the section 2.1. Plant material.

  1. Too little information about the machine learning introduction.

RESPONSE: The introduction was extended.

  1. In the discussion, different methods for processing hyperspectral reflectance data were mentioned, and it is suggested to supplement them in the test methods.

Some clarifications were added to the M&M section and the processing methods used in the paper are covered in their entirety in subsections 2.3.2 (classifiers), 2.3.3 (vegetation indices), and 2.3.4 (LBPs).

Reviewer 2 Report

In this work, the authors propose a way to classify damage of apple by spectrum features and random forest. This work is of importance to applications of machine learning in food industry and engineering. There are some suggestions for the authors before publication of the paper.

1.       Why not deep learning? It seems natural to me that convolutional neural network should be a default machine learning model to try for this work. The authors should provide a convincing discussion about why CNN is not a good choice, or provide numerical comparisons.

2.       In addition to random forest, why not other classic statistical learning models, such as logistic regression or SVM? Again, the authors should provide a convincing discussion or provide numerical comparisons.

3.       Although the authors provide the details of the RF model used in the paper, the authors should also provide the details of hyper-parameter tuning, like the search space and sampling strategy.

4.       The authors should provide a figure to better illustrate the machine learning framework: from the format of input raw images, to pre-processing, to machine learning architectures.  

5.       The authors describe that two human experts are employed to label the data. However, there are about 3% data that have different labels from human. The author should discuss more about these 3% data to gain more insights about the nature of the problem of apple damage. For example, do the 3% data lie in the decision boundary?

6.       Are the RF classifier stable with the labelling of the 3% data? In other words, if the authors use different labels for the 3%, how much would the RF classifier change?

7.       In line 123, how is the down-sampling conducted?

8.       In Table 2, the accuracies larger than 97.7% are meaningless. The authors should mention that.

9.       In line 236-237, the authors mention that accuracy is not suitable to measure the success, while the F-score shows the failure. Why accuracy and F-score have different effects? Because of imbalanced labels?

Author Response

We are grateful for the thorough and insightful evaluation of our work. Below are our detailed point-by-point responses to the comments made and questions raised by the reviewer. Specific amendments made to the manuscript are highlighted using Tracking Changes mode in the manuscript itself.

  1. Why not deep learning? It seems natural to me that convolutional neural network should be a default machine learning model to try for this work. The authors should provide a convincing discussion about why CNN is not a good choice, or provide numerical comparisons.

RESPONSE: The introductory section has been expanded to provide more rationale for opting for classic models. In brief, it is expected that a CNN would provide better numerical results at the cost of higher computational complexity; however, the goal of the paper is not to achieve SOTA performance of the model itself, but rather to study the impact of input features. Moreover, we strived to find an approach potentially usable in real-life fruit sorter which have to process many dozen fruits per second. In effect, this means working in real-time, so the lightweight approach is of utter importance. In hyperspectral remote sensing, researchers are long grappling with the gargantuan size of deep networks and have to resort to e.g. reducing the spatial reception field (like SpectralFormer) while trying to learn spectral features. We believe that CNNs will eventually be a good fit for the problem of damage detection overall, and being better equipped to make e.g. hardware choices to leverage spectral information efficiently would, inter alia, enable researchers and industry experts to use more powerful deep learning architectures. We hope that the expanded introduction conveys that stance well.

  1. In addition to random forest, why not other classic statistical learning models, such as logistic regression or SVM? Again, the authors should provide a convincing discussion or provide numerical comparisons.

RESPONSE: The reason is largely the same as above: the focus is on the comparative results, not the absolute performance of the models. That being said, we agree that SVM results provide a valuable point of comparison as they behave differently under input transformation, and have decided to provide them as well. See also our comments above.

  1. Although the authors provide the details of the RF model used in the paper, the authors should also provide the details of hyper-parameter tuning, like the search space and sampling strategy.

RESPONSE: The Materials and Methods section has been expanded accordingly (subsection 2.3.2).

  1. The authors should provide a figure to better illustrate the machine learning framework: from the format of input raw images, to pre-processing, to machine learning architectures. 

RESPONSE: A figure illustrating the framework was added (current Fig. 2).

  1. The authors describe that two human experts are employed to label the data. However, there are about 3% data that have different labels from human. The author should discuss more about these 3% data to gain more insights about the nature of the problem of apple damage. For example, do the 3% data lie in the decision boundary?

RESPONSE: Thank you for the suggestion, not including this analysis was an oversight on our part. No, corresponding pixels generally do not lie in the decision boundary, and this highlights an extremely important problem of applicability of machine learning methods in agriculture: the issue often lies not with the model itself, but with label noise. In our case, both the classifier inability to learn certain types of damage and high discrepancy between annotations were relevant. In particular, expert #2 was significantly more prone to labeling low-confidence regions as “damage” than expert #1 and, unlike some ML tasks, a "clean" set of labels might be outright impossible to produce unless the problem is considered in a broader context. Creating a separate class for low-confidence classification outcomes would largely address this issue, but this approach has very limited use in high-throughput scenarios such as industrial sorting lines. We have covered this topic in the Discussion section already, but directly addressing the relationship between the classifier decision boundary and discrepancies in human labels certainly strengthens this point, thank you. Relevant considerations were added to the Discussion.

  1. Are the RF classifier stable with the labelling of the 3% data? In other words, if the authors use different labels for the 3%, how much would the RF classifier change?

RESPONSE: We are very grateful for this question. Initially, a preliminary analysis was conducted to establish that this does not change the key conclusions of the paper and it was decided to omit the full comparison. However, following your inquiry we were able to demonstrate that the classifier is highly stable with respect to training label change (kappa=0.99). The practice of using multiple annotations for the same data set when human labels are suspected to be unreliable or uncertain is commonly not being observed, and we view this as a major impediment for practical applications of deep learning in smart agriculture.

  1. In line 123, how is the down-sampling conducted?

RESPONSE: Information on sampling is provided in subsection 2.3.2, lines 128-129 in the original version of the manuscript (lines 144-147 in the revised version). We have expanded this section a little bit.

  1. In Table 2, the accuracies larger than 97.7% are meaningless. The authors should mention that.

RESPONSE: A clarification was added to the Results section, Discussion was also expanded accordingly. We consider these accuracies to not be entirely meaningless, however: they are commonly indicative of overfitting, and studying them in more detail – in part thanks to your suggestions – has given insights about the nature of classification errors. As a shorthand for “are not indicative of a better performance” – yes, surely. A corresponding statement was added to the Results section.

  1. In line 236-237, the authors mention that accuracy is not suitable to measure the success, while the F-score shows the failure. Why accuracy and F-score have different effects? Because of imbalanced labels?

RESPONSE: Indeed, as is common in anomaly detection problems, classes are highly imbalanced. This can be clearly seen from the LBP-only classification with RFs, which does not produce any labels for the “damage” class at all: the accuracy is still at around 90%, kappa tanks to 0.65 from ~0.9-0.95 for other classifiers, and F-score drops to 0. These numbers can serve as a reference when assessing other numerical performances. We have added more of relevant discussion to the work.